# Genetic Characteristic and RNA-Seq Analysis in Transparent Mutant of Carp–Goldfish Nucleocytoplasmic Hybrid

**DOI:** 10.3390/genes10090704

**Published:** 2019-09-12

**Authors:** Lingling Zhou, Hongwei Liang, Xiaoyun Zhou, Jingyi Jia, Cheng Ye, Qiongyao Hu, Shaohua Xu, Yongning Yu, Guiwei Zou, Guangfu Hu

**Affiliations:** 1Hubei Provincial Engineering Laboratory for Pond Aquaculture, College of Fisheries, Huazhong Agricultural University, Wuhan 430070, China; 15527699872@163.com (L.Z.); zhouxy@mail.hzau.edu.cn (X.Z.); Jiajy94@163.com (J.J.); yechenging@163.com (C.Y.); HQY960819@163.com (Q.H.); xsh2018308@163.com (S.X.); yyn55880@163.com (Y.Y.); 2Key Lab of Freshwater Biodiversity Conservation Ministry of Agriculture, Yangtze River Fisheries Research Institute, The Chinese Academy of Fisheries Sciences, Wuhan 430223, China; lianghw@yfi.ac.cn

**Keywords:** transparent mutant, iridophore, pigmentation, RNA-Seq, miRNA, guanine

## Abstract

In teleost, pigment in the skin and scales played important roles in various biological processes. Iridophores, one of the main pigment cells in teleost, could produce silver pigments to reflect light. However, the specific mechanism of the formation of silver pigments is still unclear. In our previous study, some transparent mutant individuals were found in the carp–goldfish nucleocytoplasmic hybrid (CyCa hybrid) population. In the present study, using transparent mutants (TM) and wild type (WT) of the CyCa hybrid as a model, firstly, microscopic observations showed that the silver pigments and melanin were both lost in the scales of transparent mutants compared to that in wild types. Secondly, genetic study demonstrated that the transparent trait in the CyCa hybrid was recessively inherent, and controlled by an allele in line with Mendelism. Thirdly, RNA-Seq analysis showed that differential expression genes (DEGs) between wild type and transparent mutants were mainly enriched in the metabolism of guanine, such as hydrolase, guanyl nucleotide binding, guanyl ribonucleotide binding, and GTPase activity. Among the DEGs, purine nucleoside phosphorylase 4a (*pnp4a*) and endothelin receptor B (*ednrb*) were more highly expressed in the wild type compared to the transparent mutant (*p* < 0.05). Finally, miRNA-Seq analysis showed that miRNA-146a and miR-153b were both more highly expressed in the transparent mutant compared to that in wild type (*p* < 0.05). Interaction analysis between miRNAs and mRNAs indicated that miRNA-146a was associated with six DEGs (*MGAT5B*, *MFAP4*, *GP2*, *htt*, *Sema6b*, *Obscn*) that might be involved in silver pigmentation.

## 1. Introduction

In teleost, the pigment chromophores in the skin and scales could be involved in several important biological processes, such as camouflage, thermoregulation, mate choice, mimicry, warning of toxicity, and immunity [1]. The pigments are produced by several pigment cells in fish, including melanophores, xanthophores, and iridophores [2]. Pigmentation is a very complicated process, which is regulated by a series of cellular, genetic, endocrine, environmental, and physiological factors [3]. Different pigment cells in fish display different colors due to their unique organelles. The melanin particles produced by melanocytes could absorb the incident light of a specific wavelength and make fish appear black or brown. The pigments produced by xanthophores are mainly pteridine, which can filter out certain wavelengths of light to make fish appear yellow or red [4]. In addition, iridophores could produce silver pigments, in which guanine could combine with water to form purine-containing reflecting platelets [5].

The transparent trait in fish is closely related to the development of pigment cells. Generally, most fish are transparent in their early lives, while the body in adult fish is no longer transparent due to the accumulation of pigment in skin and peritoneum. The cytoplasm of iridophores contains a large number of crystalline plates, which could reflect a certain wavelength of light and make them iridescent. The color presented by iridophores belongs to the configuration color, which is different from other kinds of pigment cells [5]. The absence of silver pigments produced by iridophores could allow light to penetrate body and make fish transparent. Previous studies have reported that zebrafish *shady*, *rose*, and *barley* transparent mutations are all associated with the lost of iridophores. Mutations in *shady* could result in a lack of iridophores throughout the whole life [6,7]. Mutations in *rose*, a gene encoding the endothelin receptor b1a (ednrb1a), could cause an adult-specific reduction of iridophores [8,9]. In addition, the transparent mutant not only shows the lost of iridophores, but also a significant decrease in the number of melanocytes.

To date, a series of genes have been reported to be involved in the determination of skin color, such as pro-opiomelanocortin (*POMC*), melanocyte stimulating hormone (*MSH*), microphthalmia-associated transcription factor (*MITF*), kit oncogene (*KIT*), tyrosinase (*TYR*), tyrosine related protein-1 (*TYRP1*), and tyrosine related protein-2 (*TYRP2*) [2,10]. In melanocytes, *MITF* could induce the biosynthesis of melanin through the up-regulation of melanogenic enzyme expression [11]. Further studies found that endothelin receptor B (*ednrb*) could induce the phosphorylation of *MITF* and its mutation could reduce the pigmentation in melanocytes [12,13]. In addition, previous studies reported that purine nucleoside phosphorylase 4a (*pnp4a*) was also involved in guanine synthesis and its mutation could cause the deficiency of iridophores [14,15]. However, most of the genetic resources involved in the color of fish remain to be discovered.

MicroRNAs (miRNAs) are a group of single-stranded, non-coding RNA molecules with an average size of approximately 22 nucleotides, which play crucial roles in numerous biological processes through promoting their target gene’s degradation or inhibiting mRNA translation. It is important for the crosstalk between miRNAs and mRNAs to steady the signal transduction and the transcriptional activities as well as maintain homeostasis in many organs [16,17,18]. In mammals, it has been reported that the miR-137 could affect body color pattern in mice [19]. In Drosophila, the loss of miR-8 could reduce pigmentation in dorsal abdomen [20]. In addition, miR-429 silencing in common carp could increase the transcript level of *Foxd3*, and influence the melanin production [21]. These studies indicate that miRNAs could also play an important role in body color formation [15]. However, little is known about the functional role of miRNA in the regulation of silver pigment synthesis.

In our previous studies, some transparent mutant (TM) individuals were found in the wild type (WT) of the carp–goldfish nucleocytoplasmic hybrid (CyCa hybrid) population [22]. In the WT, the silver pigments and melanin in the scales could block the light to pass through fish body, so they were opaque. However, there were no reflective substances in the skin and peritoneum of TM, so the internal organs of TM could be observed directly. In the present study, using the WT and TM as the model, (i) the hybridization experiment was used to further explore the genetic characteristics of the transparent trait in common carp; (ii) the RNA-Seq was used to examine the differentially expressed genes between the TM and WT; (iii) the miRNA-Seq was also recruited to detect the functional role of miRNA in the formation of pigment in common carp. This study will help us to understand the formation mechanism of the transparent trait in teleost.

## 2. Materials and Methods

### 2.1. Genetic Analysis of the Transparent Trait

Experimental CyCa hybrids were bred in the Yaowan Testing Ground of the Yangtze River Fisheries Research Institute (Hubei Province). The wild types (WTs) in CyCa hybrids (three males and three females) were used as broodstocks for crossing to produce the F1 generation (Figure 1A). After the F1 generation matured, the transparent mutants (TMs) in the F1 generation (six males and six females) were used as parents for self-crossing to produce the F2 generation (Figure 1B). In addition, three male TMs in F1 generation and three female WTs in the F0 generation were used as parents for back-crossing to produce the F2 generation, respectively (Figure 1C). Similarly, three female TMs in the F1 generation and three male WTs in the F0 generation were also recruited to do the back-crossing experiment (Figure 1D). When the phenotypes were easily distinguished, the number of WTs and TMs in the F1 and F2 generation were counted, respectively. In the present study, the common carps were used according to the protocol approved by the committee for animal use at Huazhong Agricultural University (ethical approval no. HBAC20091138; date: 15 November 2009).

### 2.2. RNA-Seq Library Construction and Illumina Sequencing

Total RNA was extracted from the whole carp fry of WTs and TMs which were two-weeks old by using a TRIzol reagent (Invitrogen, Carlsbad, CA, USA) according to the manufacturer’s instructions. To be specific, five fry were collected for one replicated group, and three replicated groups for TMs (total 15 carps) and WTs (total 15 carps) were all used for RNA extraction, respectively. A Nanodrop 2000 spectrophotometer was used to assess sample purity and RNA concentration, and the quality of RNA was analyzed by using an Agilent 2100 bioanalyzer with a RNA 6000 Nano kit (Agilent Technologies, Santa Clara, CA, USA). Only high-quality RNA samples (RIN ≥ 8) were used to construct the sequencing library. RNA-Seq transcriptome libraries were prepared by using 5 μg of total RNA following TruSeq^TM^ RNA sample preparation Kit (Illumina, San Diego, CA, USA). After quantification by TBS380, paired-end RNA-Seq sequencing libraries were sequenced with the Illumina HiSeq™ 4000 (Illumnia, San Diego, CA, USA). In this study, the cDNA libraries for TMs and WTs were constructed and sequenced in Nextomics Biosciences (Wuhan, China), respectively. The raw reads were filtered using NGSQCToolkit (v2.3.3). By removing reads containing adapters, reads containing poly-N, or low quality reads from raw data, the clean data were obtained. Using FastQC software to do quality control (QC) of clean data, if QC was qualified, clean reads were compared to the reference sequence (http://www.carpbase.org/CarpBase/download). Then, a series of subsequent analysis such as gene expression, variable shear, prediction of new transcripts, SNP detection, and gene structure optimization were conducted.

### 2.3. Small RNA Library Construction and Illumina Sequencing

A total amount of 10 μg of total RNA from WTs and TMs were used as input material to construct small RNA libraries. The small RNA libraries were constructed by using a Truseq^TM^ Small RNA sample prep Kit (San Diego, CA, USA), following the Illumina protocol. Subsequently, SE50 sequencing of qualified sequencing libraries was performed on an Illumina HiSeq™ system [23]. In this study, three small RNA libraries for WTs and TMs were constructed and sequenced by Nextomics Biosciences (Wuhan, China), respectively. Then, a series of subsequent analysis such as sequencing data quality control, small RNA annotation analysis, and miRNA deep analysis would be carried out.

### 2.4. Differentially Expressed Genes (DEGs) Analysis and Functional Enrichment

To identify DEGs between TMs and WTs, the expression level of each transcript was calculated according to the fragments per kilobase of exon per million mapped reads (FRKM) method. The significance of DEGs between TMs and WTs was determined by using edgeR [24]. In this study, the threshold for DEGs was defined with *p* < 0.05 and |log2(Fold change)| ≥ 1. Functional annotation of gene ontology (GO) terms were analyzed by using Blast2GO software [25], and GO functional classification of unigenes was analyzed by using WEGO software [26]. Functional enrichment analysis including GO and Kyoto Encyclopedia of Genes and Genomes (KEGG) were performed by using Goatools (or KOBAS) software.

### 2.5. Differentially Expressed miRNAs (DEMs) Analysis and Functional Enrichment

Clean reads were filtered from raw reads by removing low quality reads and adaptor sequences, either shorter than 18 nt or longer than 32 nt. Then, the cleaned sRNA reads were mapped to the Rfam databases to discard rRNA-, snRNA-, scRNA-, snoRNA-, tRNA-, and ribozyme associated reads [27]. Then, the qualified clean reads were compared with the reference sequence by the genome comparison analysis software Bowtie (version 1.1.2) [28]. DEMs analysis was performed by using DEGseq software. The screening conditions for DEMs were *p*-adjust < 0.05 and |log2(Fold change)| ≥ 1. The potential known and novel miRNA targets were predicted by miRanda software [29], and expression levels of target genes were taken from DEGs identified in our transcriptome data. GO and KEGG analysis were performed for the target genes of DEMs by using Blast2GO and KASS.

### 2.6. Interaction Analysis of DEGs and DEMs

In order to define all the possible miRNA-mRNA interactions, we constructed the DE genes pool (*p* < 0.05 and |log2(Fold change)| ≥ 1) and DE miRNAs target genes pool (*p*-adjust < 0.05 and |log2(Fold change)| ≥ 1), respectively. Then the interaction analysis was carried out in the two gene pools. Briefly, the target genes of the DE miRNAs were firstly predicted basing on the miRanda (3.3a). Then, overlapped with the DE genes obtained by RNA-Seq, the target DE genes of DE miRNAs were further detected. According to the expression trend, the relationship between DE miRNAs and their corresponding target DE genes was determined to be up–up, up–down, down–down and down–up, respectively. The miRNA–mRNA regulatory networks were visualized by using Cytoscape software [30]. Given that miRNAs could negatively regulate the expression of their target mRNAs in most cases, only the data with a negative relationship (up–down and down–up) between miRNA and mRNA expression were selected for the miRNA–mRNA network mapping in the present study.

### 2.7. RT-PCR Analysis of DEGs

Total RNA was extracted from the same period carp fry as in the RNAseq experiment by using Trizol reagent (Invitrogen). Five fry were collected for one replicated group, and three replicated groups for TMs (total 15 carps) and WTs (total 15 carps) were all used for RNA extraction, respectively. The cDNA was synthesized by using 5 μg total RNA with the Hifair^TM^ II 1st Strand cDNA Synthesis Kit following the manufacturer’s protocol (Yesen, Shanghai). The primers of candidate genes were designed following the general principles with Premier 6.0 software. The housekeeping gene *β-actin* and glyceraldehyde-3-phosphate dehydrogenase (*Gapdh*) were used as the reference. The details of the primers used in RT-qPCR are listed in Appendix A. RT-PCR was performed by using the Unique Aptamer^TM^ qPCR SYBR^®^ Green Master Mix (Novogene, Tianjin, China) on an ABI PRISM 7500 Real-time Detection System (Life Technologies, Carlsbad, CA, USA) according to the manufacturer’s protocol. The relative expression levels of target genes were calculated by the 2^−∆∆CT^ method. ∆CT_target gene_ = CT_target gene_ − CT_housekeeping gene_, ∆∆CT = ∆CT_TM_ − ∆CT_WT_, each target gene mRNA expression difference multiples with 2^−∆∆CT^.

## 3. Results

### 3.1. Genetic Analysis of Transparent Trait

In the CyCa hybrid carp inbred line, some individuals were found to present the transparent trait without reflective substances in their eyelids and scales, which were named as transparent mutants (TM). Since the TMs accounted for about 25% in F1 generation (Figure 1A), it was presumed that the reflective trait in their broodstock was an “Aa” heterozygote, and the transparent trait was recessive over the reflective trait. In addition, all of the self-crossing F2 generation showed the transparent trait (Figure 1B), while the ratio of TMs and WTs of back-crossing F2 generation was nearly 50% (Figure 1C,D). These results further confirmed that the transparent trait in TM was “aa” and the heritance characteristic was consistent with Mendel’s genetic law. Microscopic examination showed that there were a large number of melanin, yellow pigments, and silver pigments on the scales of WT (Figure 2A). However, only a large number of yellow pigments, very few melanin and silver pigments, were found on the scales of TM (Figure 2B). Due to the absence of silver pigments, the scales in TMs were transparent.

### 3.2. Transcriptome Analysis

After stringent quality assessment and data filtering, approximately 63,272,006 and 75,019,864 clean paired-end sequence reads with a length of 145 bp and a Q30 rate (those with a base quality greater than 20) over 96% were generated from TMs and WTs, respectively (Appendix A). Compared with the reference genome of common carp, approximately 41,800,444 and 50,183,848 clean reads were mapped back to the transcriptome assembly in TM and WT libraries, respectively (Appendix A). Compared to WT, a total of 331 transcript-derived unigenes were up-regulated while 792 unigenes were down-regulated in TM (Appendix A). Here, 57 up-regulated DEGs were selected on the basis of FC_TM/WT_ > 3 and 60 down-regulated DEGs were selected according to FC_WT/TM_ > 15. Among them, the up-regulated DEGs in TM included epididymis-specific α-mannosidase, zinc-binding protein A33, *FAM83H*, interferon-induced very large GTPase 1, galactose-specific lectin nattectin, guanine nucleotide-binding protein subunit α-14, and others. (Table 1). For the down-regulated DEGs, they contained hemoglobin subunit α, 40S ribosomal protein S26, epididymis-specific α-mannosidase, purine nucleoside phosphorylase (*pnp4a*), purine nucleoside phosphorylase (*pnp5a*), and others (Table 2).

To identify the functional clusters and biochemical pathways of these DEGs, GO (gene ontology) enrichment analysis was performed by hypergeometric distribution testing. Based on the Nr annotation, 29,609 unigenes were assigned to 1739 GO terms. Using WEGO, 15,270 unigenes for the biological process category, 10,947 unigenes for the cellular component category, and 26,107 unigenes for the molecular function category were identified, respectively (Appendix A). Based on the analysis of DEGs and GO annotation results, GO enrichment analysis was conducted under the condition of *p* < 0.05. Here, we listed 21 GO terms for up-regulated DEGs and 24 GO terms for down-regulated DEGs, respectively. For the up-regulated DEGs, the most abundant GO terms were annotated to peptidase activity, heme binding, iron ion binding, and other terms (Figure 3A). For the down-regulated DEGs, the most abundant GO terms were annotated to molecular function, such as hydrolase activity, guanyl nucleotide binding, GTPase activity, nucleoside triphosphatase act, pyrophosphatase activity and GTP binding (Figure 3B).

KEGG (Kyoto Encyclopedia of Genes and Genomes) is the main public database for systematic analysis of the metabolic pathways in cells. To further elucidate the biological functions and interactions of these DEGs, the unigenes were mapped to the reference pathways recorded in the KEGG database. In total, 12,540 unigenes were annotated in KEGG and located to 5 known KEGG metabolic pathways, including cellular processes, environmental informational processing, genetic information processing, metabolism, organismal systems (Table 3). KEGG analysis of DEGs showed 4 different pathways, including other glycan degradation, antigen processing and presentation, microRNAs in cancer, and regulation of mitophagy-yeast (Table 4).

### 3.3. miRNA Analysis

For the miRNA sequences, 10,979,661 and 12,296,112 total raw reads were obtained from TM and WT, respectively. After quality control, 9,876,199 and 11,368,562 total clean reads were left in TM and WT for further analysis, respectively. Among them, 7,898,353 in TM and 9,385,680 in WT clean reads were perfectly matched to the reference genome. (Appendix A). To identify miRNAs in TM and WT, all of the mapped small RNA reads in the transcriptome were used as query against known and novel miRNAs in the database. In total, 54 known miRNAs and 396 novel miRNAs were obtained. Screening conditions for DEMs were *p*-adjust < 0.05 and |log2(Fold change)| ≥ 1. Firstly, we identified the expression profile of known miRNAs between TM and WT. The results indicated that only 2 differential expression known miRNAs, miR-146a and miR-153b, were highly up-regulated expression in TM compared to WT. Secondly, we identified the differential expression profile of novel miRNAs between TM and WT. The results indicated that 70 differential expression novel miRNAs were up-regulated and 125 differential expression novel miRNAs were down-regulated in TM vs. WT. Here, 2 up-regulated known DEMs, and 30 up-regulated and 30 down-regulated novel DEGs were listed according to the FC (fold change) value (Appendix A). The overall distribution of the DEMs can be inferred from the volcanic map (Appendix A).

GO enrichment analysis was performed for the target genes of the DEMs, including two known and 195 novel DEMs. For the target genes (total 490) of two known DEMs, 15 GO terms from the three ontologies of molecular function, cellular component, and biological process were selected to describe their functions. As a result, some GO terms were identified in the guanyl-nucleotide exchange factor activity, GTPase activator activity, purine nucleotide metabolic process, GTP metabolic process, and other functions (Figure 4). Additionally, similar to known DEMs, the most abundant GO terms in the 64,546 target genes of 195 novel DEMs were annotated to metal ion binding, nucleic acid binding, purine nucleotide binding, and cellular process (Figure 5).

KEGG enrichment analysis was performed for the target genes of DEMs, including known and novel DEMs. For the known DEMs, 138 target genes were assigned to 344 KEGG pathways. For the novel DEMs, 9824 target genes were assigned to 344 KEGG pathways. For the known DEMs, target genes were mostly concentrated in metabolic pathways, Rap1 signaling pathway, pathways in cancer, biosynthesis of secondary metabolites, biosynthesis of antibiotics, and other pathways (Figure 6A). For novel DEMs, target genes were mostly enriched in the mitogen-activated protein kinase (MAPK) signaling pathway, Human T-lymphotropicvirus 1 (HTLV-I) infection, oxytocin signaling pathway, axon guidance, microRNAs in cancer, and other pathways (Figure 6B).

### 3.4. Interaction Analysis Between miRNAs and mRNAs

In order to visualize the relationship between DEGs and DEMs, the RNA-Seq data and miRNA data were combined together to make the miRNA–mRNA regulatory network. For the known DEM, the up-regulated miR-146a was associated with 6 down-regulated DEGs including *MGAT5B*, *MFAP4*, *GP2*, *htt*, *Sema6b*, and *Obscn* (Table 5). In addition, miR-153b was also one of the known DEMs, but its target genes were not detected in the DEGs. For the novel miRNAs, correlation analysis was performed for 70 up-regulated miRNAs and their down-regulated target DEGs (Figure 7A). The network diagram showed that some novel miRNAs, such as Novel_294, Novel_384, and Novel_272, were associated with multiple differentially expressed genes. Conversely, some target genes, such as *TTN*, *Nlrc3*, and *NLRP*, were regulated by multiple novel miRNAs (Figure 7A). In addition, correlation analysis was also performed for 125 down-regulated miRNAs (such as Novel_212, Novel_377, and Novel_399) and their up-regulated target DEGs (such as *Nrxn2*, *GALNT17*, and *MLRP3*) (Figure 7B).

### 3.5. RT-qPCR Validation of DEGs

In order to substantiate the RNA-Seq data, eleven DEGs including endothelin receptor type B (*EDNRB*), up-regulator of cell proliferation (*URGCP*), purine nucleoside phosphorylase 4a (*PNP4a*), A kinase anchor protein 12a (*ALAP12a*), purine nucleoside phosphorylase 4a (*PNP5a*), GTPase IMAP family member 8 (*GIMAP8*), amyloid protein-binding protein 2 (*APBA2*), galactose-specific lectin nattectin (*GSLN*), histidine-rich glycoprotein (*HRG*), guanine nucleotide-binding protein subunit α-14(*GNA14*), and guanylate-binding protein 1(*GBP1*) were selected to validate their relative expression levels between TM and WT by using RT-qPCR. The results revealed a significant up-regulation in the expression of EDNRB, URGCP, PNP4a, AKAP12a, PNP4a, and GIMAP8 in WT relative to TM (*p* < 0.05). Similarly, a significant up-regulation was observed in the expression of APBP2, GSLN, HRG, GNA14, and GBP1 in TM relative to MT (*p* < 0.05) (Table 6). These results suggested that the qPCR results were generally consistent with the RNA-Seq data.

## 4. Discussions

There are three main types of pigment cells in fish, namely melanophores, xanthophores, and iridophores [2]. Among them, iridophores could produce silver pigments to reflect light in fish scales, so the loss of silver pigments could make the fish transparent. In goldfish, the transparent mutants have been reported in China, which were named as crystal goldfish. Further genetic studies on crystal goldfish indicated that the transparent mutants were homozygotes of guanophore (-/-) [31]. Similarly, our present study demonstrated that the transparent trait in common carp was recessive over the reflective trait, and this phenomenon could completely conform to the Mendelian segregation rule of alleles. These results suggested that the transparent common carp could be a novel model to investigate the genetic basis of silver pigment synthesis.

In the present study, microscopic observations showed that the silver pigments and melanin were both lost in the scales of transparent mutants compared to that in wild types. Furthermore, GO and KEGG analysis showed that the DEGs between wild type and transparent mutants were mainly enriched in the metabolism of guanine, such as hydrolase, guanyl nucleotide binding, guanyl ribonucleotide binding, and GTPase activity. Consistent with these, previous studies have shown that the guanine was the major component of silver pigments, and it could combine with water to form a crystal plate to make scales reflect light [32]. The reduction in guanine deposition in iridophores could induce the establishment of the transparent medaka strains [33,34] (Wakamatsu et al., 2001; Ohshima et al., 2013). In addition, the differentiation of melanophores is coupled closely with the onset of pteridine synthesis which starts from GTP and is regulated through the control of GTP cyclohydrolase I activity [14].

Transcriptome analysis showed that several DEGs were detected between TMs and WTs, including purine nucleoside phosphorylase 4a (*pnp4a*) and endothelin receptor B (*ednrb*). As a member of *pnp* gene family in vertebrates, *pnp4a* was reported to have an important role in guanine synthesis in iridophores [14]. The previously reported zebrafish *shady*, *rosy*, and *barley* transparent mutations were all associated with the deficiency of iridophores caused by the mutant of *pnp4a* [7,32,35]. The CRISPR/CAS9 system has been used to further confirm that *pnp4a* was a causal gene on the guanineless locus of zebrafish [14,15]. In the present study, the slivery pigments in TM were completely lost, and *pnp4a* was less expressed in TM compared to WT. These results, taken together, suggested that *pnp4a* could also be involved in the formation of silver pigments in common carp. Similar to *pnp4a*, our results found that the *pnp5a* was also differentially expressed between WT and TM, which indicated that the *pnp5a* might be a guanine synthetic gene in teleost iridophores. Endothelin receptor b (*ednrb*) and its ligand, endothelin (*edn*), were vital in melanocyte development [36]. The signaling pathway of *ednrb* could promote the proliferation and differentiation of melanocyte stem cells and the regeneration of epidermal melanocytes, and *ednrb* mutations could give rise to pigmentation defects [12,13]. Further studies found that the activation of *ednrb* could induce the phosphorylation of microphthalmia-associated transcription factor (*MITF*), which was the key transcription factor for catalyzing tyrosine synthesis into melanin in melanocytes [15]. In our present study, the melanin in TM was completely lost and the expression level of *ednrb* in TM was significantly lower than that in WT, which suggested that *ednrb* might play an important role in the synthesis of melanin in common carp. Guanine nucleotide-binding protein was the subunit of the G protein-couple signaling pathway, and a previous study reported that it might be involved in the formation of coat color in mice [37]. Interestingly, our present study also found that guanine nucleotide-binding protein subunit α-14 (*GNA14*) was significantly differentially expressed between WT and TM, which suggested that *GNA14* might be involved in the formation of pigment in teleost.

In mammals, miRNAs played pivotal roles in a variety of developmental processes, and their dysregulations were linked to several skin pigment diseases, such as melisma development [38] and melanoma [39]. In teleost, previous studies have reported that miR-429 was highly expressed in red skin of common carp, and inhibition of miR-429 could cause a substantial decrease in skin pigmentation [21], which suggested that the miRNAs also played an important role in the pigmentation of teleost. In our present study, DEMs analysis found that miR-146a and miR-153b were both more highly detected in TM compared to those in WT. In mammals, recent studies have reported that miR-146a played an important role in melanoma [39,40,41] and retinal pigment epithelial cells [42]. Similarly, miR-153 could suppress cell proliferation and invasion by targeting snail family transcriptional repressor 1 (SNAI1) in melanoma [43]. In the present study, further GO and KEGG analysis showed that the target genes of miR-146a and miR-153b were mainly enriched in the guanyl-nucleotide exchange factor activity, GTPase activator activity, purine nucleotide metabolic process, and GTP metabolic process. These results suggested that miR-146a and miR-153b might also play a vital role in the regulation of pigment synthesis in common carp. In addition, we also found several novel DEMs between TMs and WTs, such as Novel_294, Novel_384, Novel_272, Novel_377, Novel_212, and Novel_399. Interaction analysis showed that these DEMs were associated with several DEGs between TMs and WTs. Further GO and KEGG analysis showed that these target DEGs were mainly enriched in purine nucleotide binding and nucleic acid binding. These results suggested that these novel DEMs might also play an important role in the regulation of pigmentation.

In the present study, the transparent mutants were detected in the CyCa hybrid carp. Microscopic examination showed that the scales of transparent mutants were completely devoid of silver pigments and melanin. Genetic analysis indicated that the transparent trait was recessive over the opaque trait in common carp. To further examine the molecular mechanism of the transparent trait, RNA-Seq and miRNA-Seq were both used to detect the DEGs and DEMs between TMs and WTs. RNA-Seq analysis showed that several DEGs were detected between TMs and WTs, such as *pnp4a* and *ednrb*. Interestingly, previous studies have reported that the *pnp4a* and *ednrb* played an important role in the synthesis of silver pigments and melanin, respectively. In addition, our GO and KEGG analysis showed that the DEGs between TM and WT were mainly enriched in the metabolism of guanine, which was consistent with the previous studies that the silver pigment was the metabolic product of guanine. By using miRNA-Seq analysis, several known and novel DEMs were also detected between TM and WT. In addition, interaction analysis showed that the DEMs were associated with several DEGs. These results suggested that the miRNAs were also involved in carp pigmentation. In general, the present study could provide some new ideas for investigating the genetic mechanism of the transparent trait and the formation mechanism of sliver pigment in common carp.

## Figures and Tables

**Figure 1 genes-10-00704-f001:**
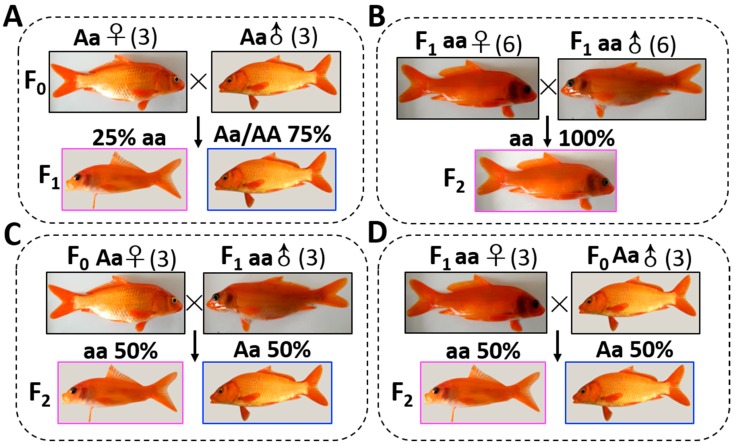
Genetic analysis of transparent trait. (**A**) The wild types (WTs) in the carp–goldfish nucleocytoplasmic hybrids (CyCa hybrids) were used as broodstocks for crossing to produce F1 generation. (**B**) After the F1 generation matured, the transparent mutants (TMs) in the F1 generation were used as parents for self-crossing to produce the F2 generation. (**C**) TMs in the F1 generation and WTs in the F0 generation were used as the male parent and female parent for back-crossing to produce the F2 generation, respectively. (**D**) TMs in the F1 generation and WTs in the F0 generation were used as the female parent and male parent for back-crossing to produce the F2 generation, respectively. When the phenotypes were easily distinguished, the number of WTs and TMs in the F1 and F2 generation were counted, respectively.

**Figure 2 genes-10-00704-f002:**
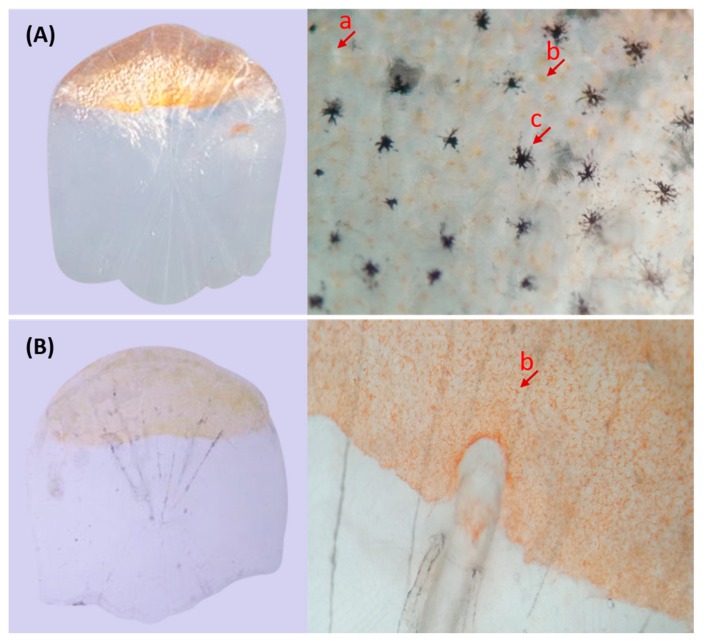
Microscopic observation of scales. (**A**) Scales of WT; (**B**) Scales of TM. a: sliver pigment; b: yellow pigment; c: melanin

**Figure 3 genes-10-00704-f003:**
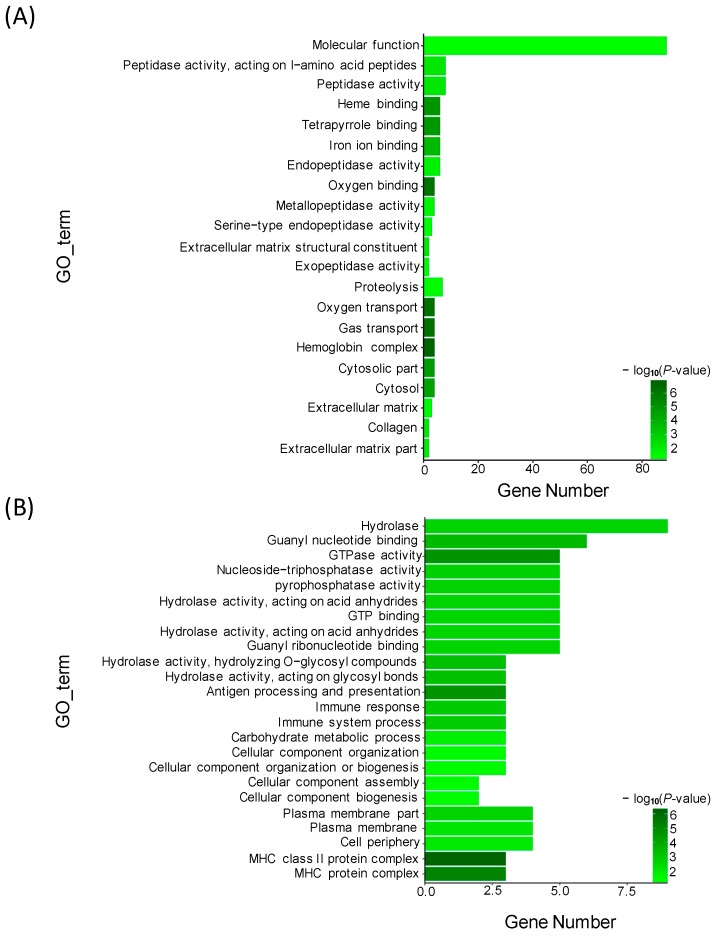
Gene ontology (GO) enrichment analysis of DE genes between TM and WT. (**A**) Significantly enriched GO terms of up-regulated DE genes; (**B**) Significantly enriched GO terms of down-regulated DE genes.

**Figure 4 genes-10-00704-f004:**
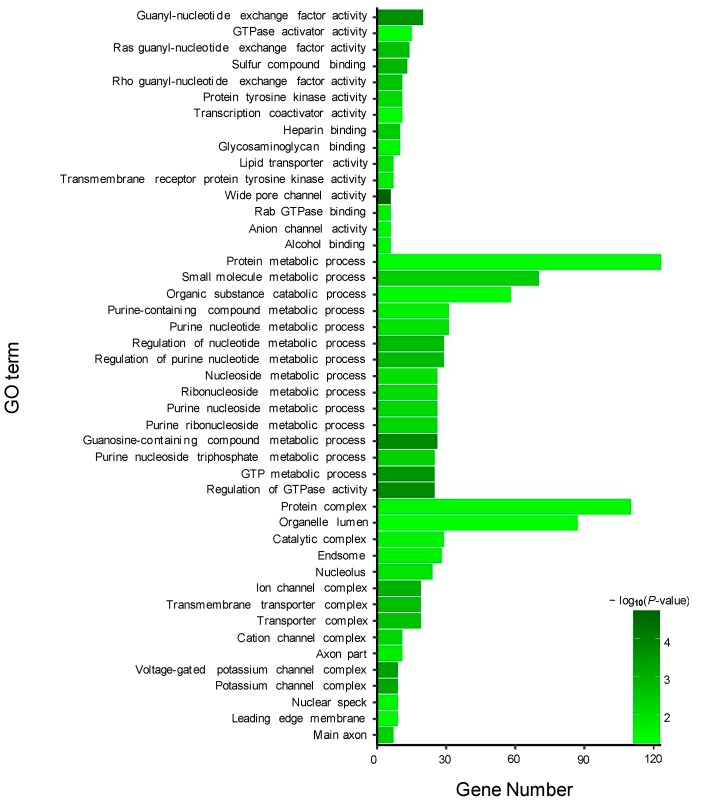
Gene ontology (GO) enrichment analysis of target genes of two known DE miRNAs between TM and WT.

**Figure 5 genes-10-00704-f005:**
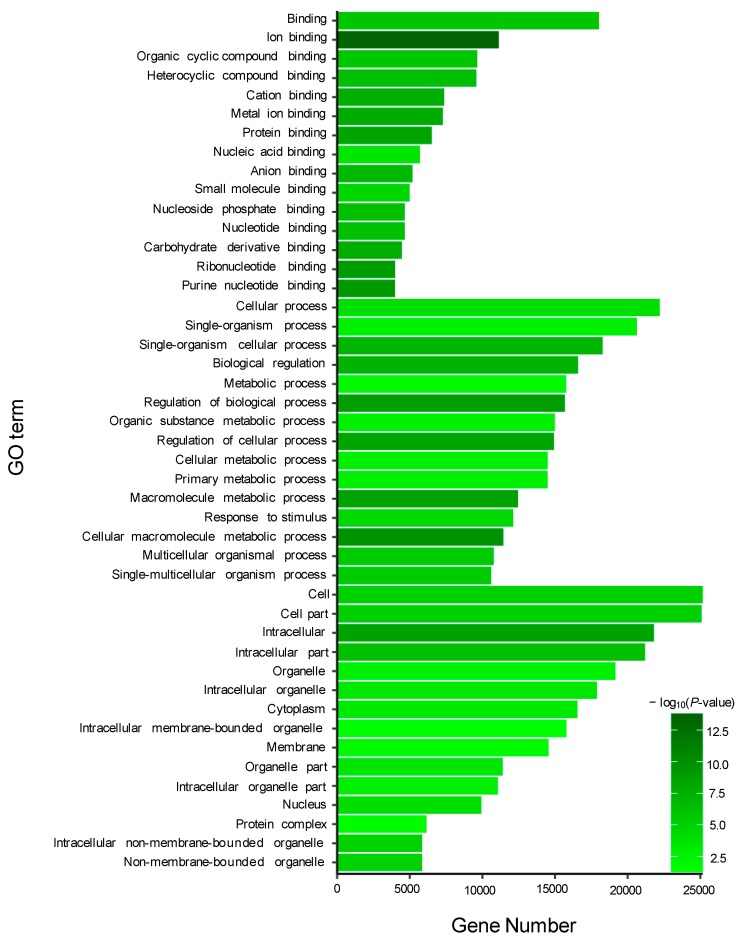
Gene ontology (GO) enrichment analysis of target genes of 195 novel DE miRNAs.

**Figure 6 genes-10-00704-f006:**
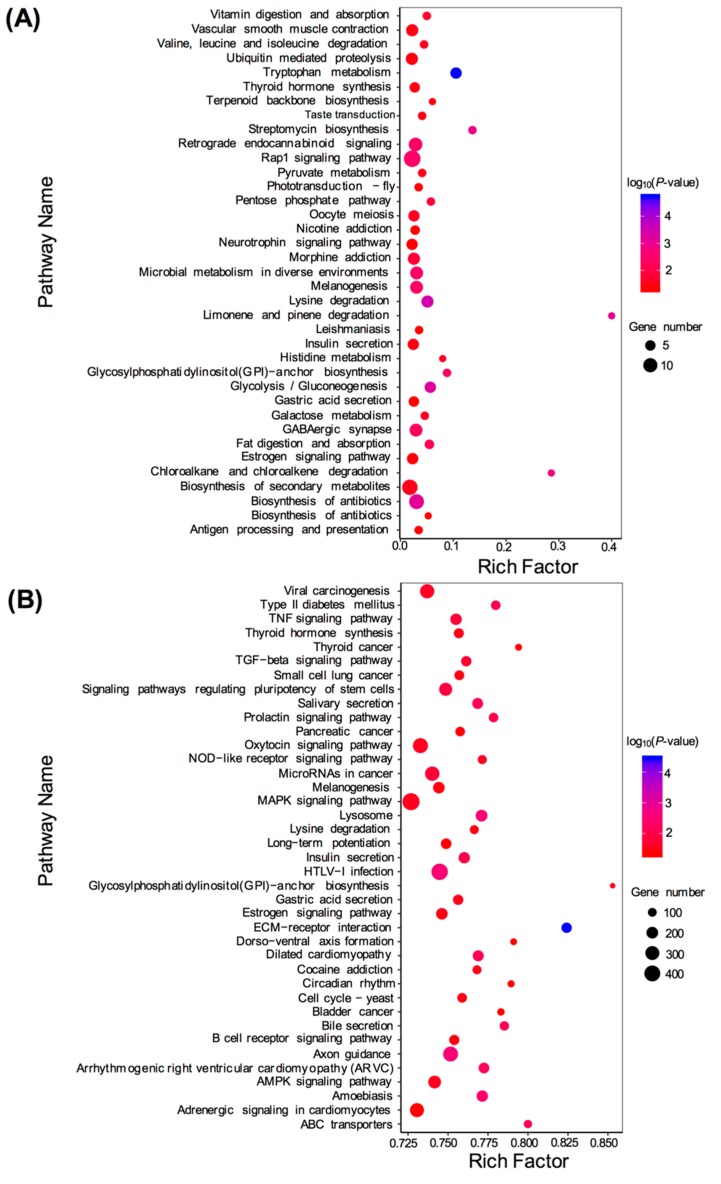
Kyoto Encyclopedia of Genes and Genomes (KEGG) pathway enrichment analysis of the target genes of differentially expressed miRNAs (DEMs). (**A**) Significantly enriched pathways of the target genes of known DEMs (TM vs. WT); (**B**) Significantly enriched pathways of the target genes of novel DEMs (TM vs. WT).

**Figure 7 genes-10-00704-f007:**
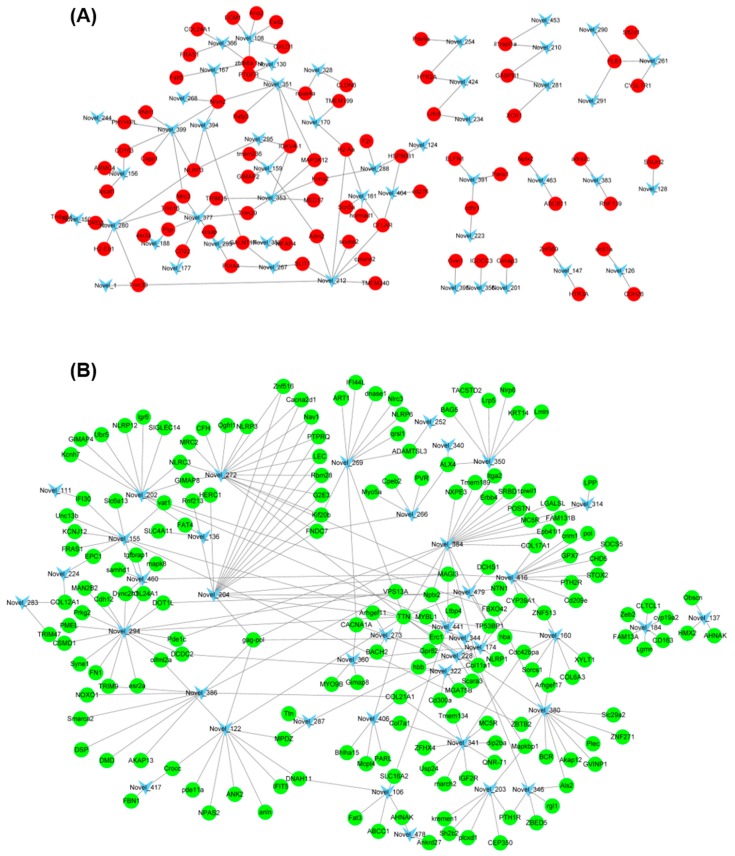
Interaction analysis of novel DE miRNAs and their target DE genes. (**A**) Interaction analysis of down-regulated novel DE miRNAs and their up-regulated target genes (TM vs. WT); (**B**) Interaction analysis of up-regulated novel DE miRNAs and their down-regulated target genes (TM vs. WT).

**Table 1 genes-10-00704-t001:** Up-regulated differential expression (DE) genes in TM compared to WT.

Gene_ID	FC	*p*-Value	Description
G_083507	38.00	2.69 × 10^−5^	galaxin-like
G_051430	26.00	4.53 × 10^−5^	zinc-binding protein A33-like; E3 ubiquitin-protein ligase TRIM39-like
G_004515	24.60	5.79 × 10^−5^	relaxin-3 receptor 1-like
G_019859	19.76	2.18 × 10^−4^	FAM217B-like; tubulin polyglutamylase TTLL9
G_020312	18.00	2.29 × 10^−3^	potassium voltage-gated channel subfamily V member 1-like
G_066161	17.37	5.30 × 10^−4^	Wilms tumor protein homolog
G_048206	16.83	5.30 × 10^−4^	PI-PLC X domain-containing protein 1-like
G_090918	16.00	7.27 × 10^−4^	germ cell-specific gene 1-like protein
G_077414	15.87	7.27 × 10^−4^	protein phosphatase 1L-like
G_018305	15.72	7.27 × 10^−4^	von Hippel-Lindau disease tumor suppressor-like; protein lifeguard 4-like
G_046961	15.51	7.27 × 10^−4^	interferon-induced very large GTPase 1-like
G_031705	14.45	2.00 × 10^−3^	histone acetyltransferase KAT7-like; protachykinin-like
G_037811	14.43	4.81 × 10^−4^	E3 ubiquitin-protein ligase RNF19B-like
G_079684	13.10	6.97 × 10^−7^	uncharacterized
G_037590	13.00	9.91 × 10^−5^	RLA class II histocompatibility antigen; MHC class II α chain
G_007981	13.00	2.86 × 10^−3^	5-hydroxytryptamine receptor 3A-like
G_025884	12.23	4.17 × 10^−3^	uncharacterized
G_014825	12.00	4.17 × 10^−3^	SLAM family member 9-like
G_039387	12.00	4.17 × 10^−3^	RLA class I histocompatibility antigen; α chain 11/11-like
G_041891	12.00	4.17 × 10^−3^	immunoglobulin superfamily DCC subclass member 3-like
G_067118	12.00	4.17 × 10^−3^	oxidoreductase-like domain-containing protein 1
G_080200	11.65	4.17 × 10^−3^	proteinase-activated receptor 1-like
G_041476	10.31	7.95 × 10^−6^	epididymis-specific α-mannosidase-like
G_088398	10.00	1.42 × 10^−3^	HMG domain-containing protein 3-like
G_016786	10.00	4.55 × 10^−3^	transcription elongation regulator 1-like protein
G_083267	8.80	3.12 × 10^−3^	guanylate-binding protein 6-like
G_073465	8.33	3.83 × 10^−3^	guanine nucleotide-binding protein subunit α-14-like
G_052838	8.00	4.73 × 10^−3^	metastasis-associated protein MTA3-like
G_045720	7.94	4.22 × 10^−4^	WD40 repeat-containing protein SMU1-like
G_075896	7.35	1.48 × 10^−4^	FAM83H-like
G_010555	6.71	1.62 × 10^−3^	carcinoembryonic antigen-related cell adhesion molecule 3-like
G_027065	6.63	1.38 × 10^−3^	slit homolog 1 protein-like; slit guidance ligand 1 (slit1)
G_043843	6.40	4.75 × 10^−3^	uncharacterized
G_064197	5.96	4.50 × 10^−4^	amyloid protein-binding protein 2-like
G_006629	5.57	4.64 × 10^−3^	neutral α-glucosidase AB-like
G_037589	5.57	4.64 × 10^−3^	H-2 class II histocompatibility antigen; I-E β chain-like
G_062240	5.22	4.51 × 10^−3^	CCAAT/enhancer-binding protein β-like
G_005497	5.20	3.99 × 10^−3^	cholesterol 25-hydroxylase-like protein 2
G_059247	5.02	3.22 × 10^−3^	ras-related protein Rab-19-like
G_036378	5.00	1.53 × 10^−3^	microfibril-associated glycoprotein 4-like
G_012543	4.91	3.22 × 10^−3^	SLIT and NTRK like family member 4 (slitrk4),
G_011657	4.90	2.75 × 10^−3^	coiled-coil domain-containing protein 149-B-like
G_001754	4.84	1.01 × 10^−3^	sucrase-isomaltase; intestinal-like
G_021086	4.72	1.59 × 10^−3^	haptoglobin-like
G_002759	4.51	1.22 × 10^−3^	H-2 class II histocompatibility antigen; MHC class II antigen β chain
G_086333	4.47	3.30 × 10^−3^	dynein regulatory complex subunit 7; coiled-coil domain containing 135
G_077730	4.46	1.75 × 10^−3^	tripartite motif-containing protein 16-like
G_016206	4.44	2.67 × 10^−3^	guanylate-binding protein 1-like
G_016103	4.39	1.05 × 10^−3^	ladderlectin-like; galactose-specific lectin nattectin-like
G_030062	4.37	1.06 × 10^−3^	galactose-specific lectin nattectin-like; ladderlectin-like
G_002737	4.23	4.28 × 10^−3^	guanylate-binding protein 1-like
G_089784	4.19	4.28 × 10^−3^	uncharacterized
G_012483	4.08	2.30 × 10^−3^	uncharacterized; zinc finger MYM-type protein 1-like
G_009580	4.02	2.73 × 10^−3^	centriole, cilia and spindle-associated protein-like
G_039702	3.74	3.28 × 10^−3^	insulin-like growth factor-binding protein 1
G_046183	3.64	4.95 × 10^−3^	titin-like; SLAM family member 5-like
G_030093	3.55	4.29 × 10^−3^	histidine-rich glycoprotein-like; antihemorrhagic factor cHLP-B-like

FC: fold change.

**Table 2 genes-10-00704-t002:** Down-regulated DE genes in TM compared to WT.

Gene_ID	FC	*p*-Value	Description
G_066699	364.96	1.62 × 10^−14^	hemoglobin subunit α-like
G_048410	351.72	8.64 × 10^−17^	40S ribosomal protein S26
G_029206	300.97	1.82 × 10^−17^	hemoglobin subunit α-like
G_011161	289.54	8.18 × 10^−16^	hemoglobin subunit β-like
G_051354	172.25	1.93 × 10^−11^	epididymis-specific α-mannosidase-like
G_068758	138.76	1.29 × 10^−10^	toll-like receptor 12
G_029207	111.89	1.34 × 10^−13^	β-globin
G_022355	53.02	7.49 × 10^−9^	epithelial cell adhesion molecule-like (EP-CAM)
G_072107	52.99	4.66 × 10^−7^	uncharacterized
G_057519	49.02	8.76 × 10^−7^	NACHT, LRR and PYD domains-containing protein 3-like (NLRP3)
G_025896	43.65	2.05 × 10^−6^	storkhead box 2 (stox2)
G_031223	37.92	5.19 × 10^−6^	sterile α motif domain-containing protein 3-like (SAMD3) Smad3
G_070770	36.00	5.69 × 10^−5^	titin-like
G_027993	35.58	5.69 × 10^−5^	carcinoembryonic antigen-related cell adhesion molecule 1
G_065676	32.17	2.15 × 10^−7^	nectin-1-like
G_047452	32.00	1.27 × 10^−4^	SLAM family member 9-like
G_077579	32.00	1.27 × 10^−4^	parathyroid hormone/parathyroid hormone-related peptide receptor-like
G_077061	30.67	2.25 × 10^−5^	mucin-17-like; microtubule-associated protein 1B-like; melanoma 1 protein
G_073986	30.00	2.82 × 10^−5^	tetraspanin-9-like
G_012754	30.00	1.95 × 10^−4^	uncharacterized
G_018847	29.50	7.49 × 10^−6^	trypsin-1-like
G_090663	29.32	3.57 × 10^−5^	protein NLRC3-like (NOD-like receptors, NLRs)
G_089118	28.00	4.53 × 10^−5^	sterile α motif domain-containing protein 3-like
G_086324	26.70	5.79 × 10^−5^	Palmitoyl transferase ZDHHC18-like
G_060152	26.00	7.45 × 10^−5^	Gig2-like protein CauA
G_053162	26.00	4.88 × 10^−4^	serine/threonine-protein kinase mTOR-like
G_042078	25.00	9.64 × 10^−5^	uncharacterized
G_091658	25.00	6.22 × 10^−4^	PDZ domain-containing protein 2-like
G_004836	24.67	4.77 × 10^−6^	NACHT, LRR and PYD domains-containing protein 1-like. (NLRP1)
G_020260	24.31	1.26 × 10^−4^	U6 snRNA-associated Sm-like protein LSm7
G_040650	23.46	1.65 × 10^−4^	protein NLRC3-like; NACHT, LRR and PYD domains-containing protein 3
G_002722	23.32	1.65 × 10^−4^	prelamin A recognition factor like
G_028996	23.00	1.03 × 10^−3^	low-density lipoprotein receptor-related protein 2-like
G_091012	22.50	5.78 × 10^−5^	uncharacterized
G_032550	21.75	2.18 × 10^−4^	2’,5’-phosphodiesterase 12-like (PDE12)
G_024701	21.30	3.04 × 10^−3^	protein NLRC3-like; NACHT, LRR and PYD domains-containing protein 3
G_042665	20.94	4.20 × 10^−5^	uncharacterized; E3 ubiquitin-protein ligase CBL-like
G_068144	20.00	2.29 × 10^−3^	cyclic AMP-responsive element-binding protein 5-like
G_077583	20.00	2.29 × 10^−3^	trophoblast glycoprotein-like
G_083587	20.00	2.29 × 10^−3^	laminin subunit α-5-like
G_068323	19.79	3.91 × 10^−4^	receptor-type tyrosine-protein phosphatase F-like
G_001532	19.64	3.91 × 10^-04^	class A basic helix-loop-helix protein 15
G_031537	19.58	3.91 × 10^−4^	presenilins-associated rhomboid-like protein
G_059937	19.34	5.30 × 10^−4^	suppressor of cytokine signaling 5-like
G_058882	19.04	3.04 × 10^−3^	mitogen-activated protein kinase 8; JNK1 (JNK1)
G_035457	19.00	5.30 × 10^−4^	Chymotrypsinogen B-like
G_037554	19.00	3.04 × 10^−3^	protocadherin-16-like
G_039404	19.00	3.04 × 10^−3^	collagen α-1(XXI) chain-like
G_068856	19.00	3.04 × 10^−3^	anoctamin-5-like
G_089229	18.81	5.30 × 10^−4^	gastrula zinc finger protein XlCGF57.1-like
G_071373	18.00	4.07 × 10^−3^	γ-adducin-like; adducin 3 (add3)
G_075405	18.00	4.07 × 10^−3^	tektin 2 (tekt2)
G_081069	18.00	4.07 × 10^−3^	oogenesis-related gene (org)
G_015338	17.93	4.07 × 10^−3^	SLAM family member 7-like; CD48 antigen-like
G_050438	17.85	7.27 × 10^−4^	nuclear export mediator factor (nemf)
G_052930	17.77	3.36 × 10^−4^	SE-cephalotoxin-like
G_065717	17.52	7.97 × 10^−4^	voltage-dependent L-type calcium channel subunit α-1C-like
G_091453	17.00	1.01 × 10^−3^	sodium/calcium exchanger 2-like
G_007871	16.23	1.92 × 10^−4^	rho GTPase-activating protein 17-like
G_079472	16.06	1.41 × 10^−3^	chromodomain-helicase-DNA-binding protein 6-like

**Table 3 genes-10-00704-t003:** Kyoto Encyclopedia of Genes and Genomes (KEGG) classification of DE genes between TM and WT.

KEGG Pathway	Secondary Pathway	Number
Cellular Processes	Cell growth and death	786
Cellular Processes	Cell motility	219
Cellular Processes	Cellular community	422
Cellular Processes	Transport and catabolism	1131
Environmental Information Processing	Membrane transport	68
Environmental Information Processing	Signal transduction	1759
Environmental Information Processing	Signaling molecules and interaction	74
Genetic Information Processing	Folding, sorting and degradation	948
Genetic Information Processing	Replication and repair	235
Genetic Information Processing	Transcription	403
Genetic Information Processing	Translation	837
Metabolism	Amino acid metabolism	389
Metabolism	Biosynthesis of secondary metabolites	92
Metabolism	Carbohydrate metabolism	626
Metabolism	Energy metabolism	293
Metabolism	Glycan biosynthesis and metabolism	326
Metabolism	Lipid metabolism	545
Metabolism	Metabolism of cofactors and vitamins	222
Metabolism	Metabolism of other amino acids	209
Metabolism	Metabolism of terpenoids and polyketides	73
Metabolism	Nucleotide metabolism	361
Metabolism	Overview	357
Metabolism	Xenobiotics biodegradation	147
Organismal Systems	Aging	402
Organismal Systems	Circulatory system	343
Organismal Systems	Development	210
Organismal Systems	Digestive system	599
Organismal Systems	Endocrine system	1063
Organismal Systems	Environmental adaptation	430
Organismal Systems	Excretory system	187
Organismal Systems	Immune system	582
Organismal Systems	Nervous system	683
Organismal Systems	Sensory system	204

**Table 4 genes-10-00704-t004:** KEGG analysis of DE genes (TM/WT).

Pathway	*p*-Value	Gene Number	Gene_ID	Regulation	Description
Other glycan degradation	0.002	2	G_051354	Up	epididymis-specific α-mannosidase
			G_041476	Down	epididymis-specific α-mannosidase
Antigen processing and presentation	0.010	2	G_014100	Up	γ-interferon-inducible lysosomal thiol reductase
			G_065294	Up	legumain
MicroRNAs in cancer	0.045	2	G_053162	Up	serine/threonine-protein kinase mTOR
			G_043688	Up	DNA (cytosine-5)-methyltransferase 3A
Regulation of mitophagy-yeast	0.048	2	G_053162	Up	serine/threonine-protein kinase mTOR
			G_087121	Up	dynamin-2

**Table 5 genes-10-00704-t005:** Interaction analysis of miR-146a and its target DE genes.

miRNA_ID	miRNA Regulation	Target Gene ID	Gene Regulation	UniProt KB-AC	Gene Name
miR-146a	Up	G_010841	Down	Q3V5L5	*MGAT5B*
miR-146a	Up	G_019763	Down	P55918	*MFAP4*
miR-146a	Up	G_053967	Down	P25291	*GP2*
miR-146a	Up	G_058480	Down	P51112	*htt*
miR-146a	Up	G_061471	Down	O54951	*Sema6b*
miR-146a	Up	G_084436	Down	A2AAJ9	*Obscn*

**Table 6 genes-10-00704-t006:** Comparison of relative qPCR results with RNA-Seq results.

**Gene ID**	**Gene Name**	**FC_WT/TM_ (RNA-Seq)**	**FC_WT/TM_ (qPCR)**	
			***β-actin***	***Gapdh***
G_039032	endothelin receptor type B (*EDNRB*)	9.556	1.781	1.843
G_026459	up-regulator of cell proliferation (*URGCP*)	9.571	7.876	8.424
G_045698	purine nucleoside phosphorylase 4a (*PNP4a*)	12.400	6.719	6.476
G_059817	A kinase anchor protein 12a (*AKAP12a*)	8.382	2.248	2.494
G_049977	purine nucleoside phosphorylase 5a (*PNP5a*)	12.033	2.944	3.691
G_058087	GTPase IMAP family member 8 (*GIMAP8*)	9.674	2.892	2.833
**Gene ID**	**Gene Name**	**FC_TM/WT_ (RNA-Seq)**	**FC_TM/WT_ (qPCR)**	
			***β-actin***	***Gapdh***
G_064197	amyloid protein-binding protein 2 (*APBA2*)	5.962	1.679	2.352
G_030062	galactose-specific lectin nattectin (*GSLN*)	4.367	10.409	8.041
G_030093	histidine-rich glycoprotein (*HRG*)	3.547	1.613	2.614
G_073465	guanine nucleotide-binding protein subunit α-14 (*GNA14*)	8.333	8.773	7.813
G_002737	guanylate-binding protein 1 (*GBP1*)	4.232	2.678	3.745

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
