# Peer review of "Genetic Characteristic and RNA-Seq Analysis in Transparent Mutant of Carp–Goldfish Nucleocytoplasmic Hybrid"

_genes, 2019, doi:10.3390/genes10090704_

Round 1

Reviewer 1 Report

In this manuscript entitled “Genetic characteristic and RNA-Seq analysis in transparent mutant of carp-goldfish nucleocytoplasmic hybrid” (Ref: Genes: 563934) by Zhou and colleagues, the authors aimed to contribute to the investigation of the occurrence of lack of pigmentation in teleost fish. Therefore, they performed comparative expression analysis between wild type and transparent mutant fish of a carp–goldfish nucleocytoplasmic hybrid (CyCa hybrid) population using RNAseq and miRNAseq technologies. The approach of combining microscopic analysis with genetic investigations by RNAseq and miRNASeq to determine the mechanisms of silver pigmentation is good, but the methodological presentation of the experiments shows some weaknesses. In addition, the interpretation of the results is very one-sided and offers little new knowledge in the current version. For a publication in "Genes" these points should therefore be revised. 

Highlights:

Items 2 and 3 should also be extended/changed according to the revision of the manuscript

Introduction:

Lines 76 to 82: You should include the genes ednrb and pnp4a in the list here. There are, as you mention in the discussion section (lines 345 to 363), several studies suggesting their importance for the indophore pigmentation in teleost fish.

Material and Methods:

Information on the number of animals used is missing continuously throughout this part. Lines 109 to 116: Did you repeat the crossing experiment of the CyCa hybrids? RNAseq and miRNA experiments: How many animals did you use? Which tissues did you use for RNA extraction? Alternatively, did you use a tissue-mix? How old were the fish? Which gender? Can you exclude gender-effects? How about biological replicates? Line 122: The RIN values of ≥5 seem to be very low. Could you please provide the electropherograms so that one can understand the degree of degradation. Lines 127 to 132: Please include more information about the Quality Control of the raw reads and the subsequent analysis mentioned. qPCR analysis: The use of only one housekeeping gene is no longer sufficient. Did you use RNA of the same animals as in the RNAseq experiment? If I understand correctly, you used two samples and these as a triplicate? If so, this seems to me to be a much too small sample size.

Results:

lines 211, 212: Supplemental Figure 1A is not sufficient to show the different regulated unigenes. It would be better to also provide a table with all regulated unigenes in the supplement. Lines 212, 213: Which fold-change did you use as cut-off to select the 57 up- and 60 down-regulated genes? Lines 224 to 236: Details can be omitted since Supplemental Figure 2 is provided. Lines 276, Fig.4+5: You aim to identify miRNAs involved in the lack of pigmentation of the fish and therefore compare DE miRNAs. Why do you then provide GO term enrichment for all identified miRNAs (known and unknown) and not for the identified 2 DE known miRNAs and 195 DE novel miRNAs of interest? Line 306 and Table 5: Why do you only provide information for miR-146a and not for miR-153b? Lines 309 to 314: “The network diagram showed that some novel miRNAs, such as Novel_294, Novel_384 and Novel_272, were associated with multiple differential 310 expression genes. Conversely, some target genes, such as TTN, Nlrc3 and NLRP, were regulated by 311 multiple novel miRNAs”. These statements are unfortunately not apparent from Figure 7. No details are given on individual miRNAs and the genes regulated by them. Figure 7 is a general statement. In order to understand details it is necessary to have an overview of the miRNAs relevant for the pathways and the genes regulated by them. qPCR validation of DE genes: “Although there were some quantitative differences between the two analytical platforms, the similarities between the RNA-Seq data and the RT- qPCR suggest that the RNA-Seq data were reproducible and reliable.” I don`t agree with that. The validation of the expression differences of 4 of the 7 genes is insufficient (G_039032, G_059817, G_064197, G_030093). In the case of three (G_039032, G_064197, G_030093), the fold-changes determined by qPCR would not be taken into account at all due to the cut-off. Furthermore the sample size is with n=2 to small.

Discussion:

The overall problem with the discussion is that the interpretation of the results is very one-sided and offers little new knowledge. To me it is incomprehensible why the authors limit the discussion to the two genes pnp4a and ednrb as well as the miRNA-146a only. It is not explained why these genes alone should play a central role when many genes were differentially regulated. The set of DE genes should be discussed in more detail. Also, why is only miRNA-146a discussed and not the other known miRNA-153b? Could this miRNA also be involved? How about all the others? Although genes associated to miRNA-146a are mentioned there possible involvement in the occurrence of lack of pigmentation is not discussed. Please improve the whole discussion section. lines 336 to 341: This part fits better into the introduction. Line 364: replace “expression” with “expressed”

Tables and Figures

It is not clear when you refer to DE unigenes and when to DE genes. Please adjust the captions. Supplementary Table 1: Gene ID and size of the amplicon are missing. Gene “cell proliferation” - Which specific gene do you mean? Supplementary Table 1: Please include the official gene symbols using HUGO nomenclature. Furthermore, it would be good to name the NCBI gene ID to make it possible to reproduce the primer design. Figure 2: Please mark the different pigmentations.

Author Response

Responses to Reviewer 1#

Q1. Comments and Suggestions for Authors

In this manuscript entitled “Genetic characteristic and RNA-Seq analysis in transparent mutant of carp-goldfish nucleocytoplasmic hybrid” (Ref: Genes: 563934) by Zhou and colleagues, the authors aimed to contribute to the investigation of the occurrence of lack of pigmentation in teleost fish. Therefore, they performed comparative expression analysis between wild type and transparent mutant fish of a carp–goldfish nucleocytoplasmic hybrid (CyCa hybrid) population using RNAseq and miRNAseq technologies. The approach of combining microscopic analysis with genetic investigations by RNAseq and miRNASeq to determine the mechanisms of silver pigmentation is good, but the methodological presentation of the experiments shows some weaknesses. In addition, the interpretation of the results is very one-sided and offers little new knowledge in the current version. For a publication in "Genes" these points should therefore be revised.

Reponses:

Thank you very much for your help with the paper review. As you suggested, the manuscript has been revised. The responses to your comments and answers to specific questions are listed online. To facilitate the paper review, modification in the revised manuscript related to the comments/questions raised have been highlighted in yellow. Thanks a lot again.

Q2. Highlights: Items 2 and 3 should also be extended/changed according to the revision of the manuscript.

Reponses:

Thank you very much for your suggestions. Considering that this journal does not need “Highlights” section, items 1, 2 and 3 have been deleted.

Q3. Introduction: Lines 76 to 82: You should include the genes ednrb and pnp4a in the list here. There are, as you mention in the discussion section (lines 345 to 363), several studies suggesting their importance for the iridophore pigmentation in teleost fish.

Reponses:

Thank you very much for your comments. As you suggested, we have added the information for ednrb and pnp4a in the Introduction (lines 63 to 67).

Q4. Material and Methods: Information on the number of animals used is missing continuously throughout this part.

Reponses:

Thank you very much for your comments. As you suggested, the number of animals has been added in the revised manuscript (Figure 1 and lines 94 to 100).

Q5. Lines 109 to 116: Did you repeat the crossing experiment of the CyCa hybrids?

Reponses:

Thank you very much for your comments. Yes, we repeated the crossing experiment of the CyCa hybrids. The repeat number has been added in the Figure 1 and lines 94 to 100.

Q6. RNAseq and miRNA experiments: How many animals did you use? Which tissues did you use for RNA extraction? Alternatively, did you use a tissue-mix? How old were the fish? Which gender? Can you exclude gender-effects? How about biological replicates?

Reponses:

(1) Which tissues did you use for RNA extraction? Alternatively, did you use a tissue-mix? How old were the fish?

Thank you very much for your comments. In the present study, the whole carp fry, which were two-week old, were used for RNA extraction (lines 106 to 108).

(2) Which gender? Can you exclude gender-effects?

According to our back-crossing experiment (Figure 1C&D), we think the gender have no effect on the transparent trait. Since sexual dimorphism was not apparent in these fry, carps of mixed sexes were used for RNA extraction.

(3) How many animals did you use? How about biological replicates?

In the present study, five fry were collected for one replicated group. Three replicated groups for TMs (total 15 carps) and WTs (total 15 carps) were all used for RNA extraction, respectively (lines 108 to 110).

Q7. Line 122: The RIN values of ≥5 seem to be very low. Could you please provide the electropherograms so that one can understand the degree of degradation.

Reponses:

Thank you very much for your comments. Actually, the RIN values of our samples were all ≥8 (lines 112 to 113). Attached please find the electropherograms and report for our samples.

Q8. Lines 127 to 132: Please include more information about the Quality Control of the raw reads and the subsequent analysis mentioned.

Reponses:

Thank you very much for your suggestions. As you suggested, we have added more information about the Quality Control of the raw reads and the subsequent analysis mentioned (Line117-123).

Q9. qPCR analysis: The use of only one housekeeping gene is no longer sufficient. Did you use RNA of the same animals as in the RNAseq experiment? If I understand correctly, you used two samples and these as a triplicate? If so, this seems to me to be a much too small sample size.

Reponses:

Thank you very much for your comments. As you suggested, another housekeeping gene “Gapdh” was used to normalize the data (Line166-167 and Table 6 and Supplemental table 1).

Yes, we used RNA of the same period carp fry as in the RNAseq experiment to do the real-time PCR (Line160-163).

For the sample numbers in this experiment, five fry were collected for one replicated group, and three replicated groups for TMs (total 15 carps) and WTs (total 15 carps) were all used for RT-qPCR, respectively (Line160-163).

Q10. Results: lines 211, 212: Supplemental Figure 1A is not sufficient to show the different regulated unigenes. It would be better to also provide a table with all regulated unigenes in the supplement.

Reponses:

Thank you very much for your comments. Due to the large amount of data, all the differentially regulated genes have been shown in the attached Supplemental Table 5.

Q11. Lines 212, 213: Which fold-change did you use as cut-off to select the 57 up- and 60 down-regulated genes?

Reponses:

Thank you very much for your comments. 57 up-regulated DEGs were selected on the basis of FCTM/WT>3, and 60 down-regulated DEGs were selected according to FCWT/TM>15 (Line224-226).

Q12. Lines 224 to 236: Details can be omitted since Supplemental Figure 2 is provided.

Reponses:

Thank you very much for your suggestions. As you suggested, the details have been deleted in the revised manuscript.

Q13. Lines 276, Fig.4+5: You aim to identify miRNAs involved in the lack of pigmentation of the fish and therefore compare DE miRNAs. Why do you then provide GO term enrichment for all identified miRNAs (known and unknown) and not for the identified 2 DE known miRNAs and 195 DE novel miRNAs of interest?

Reponses:

I am so sorry for the confusing. In the present analysis, there were 490 target genes for the identified 2 DE known miRNAs, and 64546 target genes for the identified 195 DE novel miRNAs. Actually, in Figure 4, the GO term enrichment was used to analysis the 490 target genes of two identified DE known miRNAs. Similarly, in Figure 5, we provide GO term enrichment for the 64546 target genes of 195 identified DE novel miRNAs. We have added the description in the revised manuscript (Line279-286) and the figure legends of Fig.4 and 5.

Q14. Line 306 and Table 5: Why do you only provide information for miR-146a and not for miR-153b?

Reponses:

Thank you very much for your comments. According to our interaction analysis, miRNA-153b was one of the DE known miRNAs, but its target genes were not detected in our DE genes. This is the reason why we did not provide the information for miR-153b before. We have added an explanation about this question in the revised manuscript (Line309-310).

Q15. Lines 309 to 314: “The network diagram showed that some novel miRNAs, such as Novel_294, Novel_384 and Novel_272, were associated with multiple differential 310 expression genes. Conversely, some target genes, such as TTN, Nlrc3 and NLRP, were regulated by 311 multiple novel miRNAs”. These statements are unfortunately not apparent from Figure 7.

Reponses:

Thank you very much for your comments. I'm so sorry that I put Figure 6 in the position of Figure 7 by mistake. We have replaced it with the new Figure 7 in the revised manuscript (Figure 7). In the new Figure 7, these statements for miRNA and target genes were all apparent.

Q16. No details are given on individual miRNAs and the genes regulated by them. Figure 7 is a general statement. In order to understand details it is necessary to have an overview of the miRNAs relevant for the pathways and the genes regulated by them.

Reponses:

I'm so sorry that I put Figure 6 in the position of Figure 7 by mistake. We have replaced it with the new Figure 7 in the revised manuscript (Figure 7). The new Figure 7 clearly showed the overview of the miRNAs relevant for the pathways and the genes regulated by them.

Q17. qPCR validation of DE genes: “Although there were some quantitative differences between the two analytical platforms, the similarities between the RNA-Seq data and the RT- qPCR suggest that the RNA-Seq data were reproducible and reliable.” I don`t agree with that. The validation of the expression differences of 4 of the 7 genes is insufficient (G_039032, G_059817, G_064197, G_030093). In the case of three (G_039032, G_064197, G_030093), the fold-changes determined by qPCR would not be taken into account at all due to the cut-off. Furthermore, the sample size is with n=2 to small.

Reponses:

Thank you very much for your comments. In the present study, five fry were collected for one replicated group. Three replicated groups for TMs (total 15 carps) and WTs (total 15 carps) were all used for RNA extraction, respectively. So we think the sample size should be enough in our study (Line160-163).

Sure, compared to the RNA-seq, the fold-changes for the three genes (G_039032, G_064197, G_030093) were lower in qPCR result, but the real-time PCR results also displayed that the expression levels of the three genes were all significantly different between WT and TM.

As you suggested, to further confirm the transcriptomic results, another 2 up-regulated DE genes and 2 down-regulated DE genes were examined by using real-time PCR. (Line 324-334 and Table 6)

Q18. Discussion:

The overall problem with the discussion is that the interpretation of the results is very one-sided and offers little new knowledge. To me it is incomprehensible why the authors limit the discussion to the two genes pnp4a and ednrb as well as the miRNA-146a only. It is not explained why these genes alone should play a central role when many genes were differentially regulated. The set of DE genes should be discussed in more detail.

Reponses:

Thank you very much for your comments. As you suggested, we have added more information for other DE genes in the revised manuscript with yellow highlights. (Line 347-357; Line 364-368; Line 377-381)

Q19. Also, why is only miRNA-146a discussed and not the other known miRNA-153b? Could this miRNA also be involved? How about all the others? Although genes associated to miRNA-146a are mentioned there possible involvement in the occurrence of lack of pigmentation is not discussed.

Reponses:

Thank you very much for your comments. As you suggested, more information for the miR-153b and their target genes have been added in the revised manuscript with yellow highlights. (Line 382-384; Line 387-395; Line398-399)

Q20. Please improve the whole discussion section.

Reponses:

Thank you very much for your suggestions. As you suggested, we have try our best to improve the discussion section in the revised manuscript with highlights. (Line 338-357; Line 364-368; Line 377-381; Line 382-399).

Q21. lines 336 to 341: This part fits better into the introduction. Line 364: replace “expression” with “expressed”

Reponses:

Thank you very much for your comments. In this part, we would like to use the transparent goldfish with the homozygote of guanophore (-/-) to support our genetic results that transparent trait was recessive over reflective trait in common carp. To make it clear, we have revised this paragraph completely. (Line 338-346)

In addition, as you suggested, the “differential expression genes” has replaced with “differentially expressed genes” or “DEGs”.

Tables and Figures

Q22. It is not clear when you refer to DE unigenes and when to DE genes. Please adjust the captions.

Reponses:

Thank you very much for your comments. We have adjusted the captions in all Tables and Figures.

Q23. Supplementary Table 1: Gene ID and size of the amplicon are missing.

Reponses:

Thank you very much for your comments. As you suggested, the missing Gene ID and size of the amplicon have been added in the Supplementary Table 1.

Q24. Gene “cell proliferation” - Which specific gene do you mean?

Reponses:

Thank you very much for your comments. The correct full name is “up-regulator of cell proliferation (URGCP)”

Q25. Supplementary Table 1: Please include the official gene symbols using HUGO nomenclature. Furthermore, it would be good to name the NCBI gene ID to make it possible to reproduce the primer design.

Reponses:

Thank you very much for your suggestion. As you suggested, the HUGO names and NCBI gene ID have been added in the Supplementary Table 1.

Q26. Figure 2: Please mark the different pigmentations.

Reponses:

Thank you very much for your comments. The different pigmentations have been marked in the Figure 2.

Reviewer 2 Report

The authors describe genetic and transcriptome analyses of wild type and transparent mutant crosses of carp. For the most part, the analyses are well done and robust but there are several places that need further information and clarification (listed below). The greatest weakness of this manuscript is the English usage. The manuscript is rife with improper usage of plural and singular forms, non-capitalization of sentences, and poor syntax at times. The first bullet point in the “Highlights” section (not “Highlight”) is a good example. There are at least two common English errors in that sentence.  There are more throughout the abstract and introduction.  

Specific technical comments:

How many wild type (WT) and transparent mutants were used to extract RNA from? What were the overall numbers of fish used? Under section 2.6: How were correlation analyses carried out? (i.e. what statistical test was used to establish significant correlations?). This section needs a lot of further clarification. This is referred to again in section 3.4 but the methodology for correlation (positive and negative) is unclear. What was significant?

Author Response

Responses to Reviewer 2# 

Comments and Suggestions for Authors

Q1. The authors describe genetic and transcriptome analyses of wild type and transparent mutant crosses of carp. For the most part, the analyses are well done and robust but there are several places that need further information and clarification (listed below).

Reponses:

Thank you very much for your help with the paper review. As you suggested, the manuscript has been revised. The responses to your comments and answers to specific questions are listed online. To facilitate the paper review, modification in the revised manuscript related to the comments/questions raised have been highlighted in yellow. Thanks a lot again.

Q2. The greatest weakness of this manuscript is the English usage. The manuscript is rife with improper usage of plural and singular forms, non-capitalization of sentences, and poor syntax at times.

Reponses:

Thank you very much for your comments. Sorry for our poor English. As you suggested, we have try our best to correct the improper usage of plural and singular forms, non-capitalization of sentences, and poor syntax at times. The revised manuscript has been labeled with yellow color highlights.

Q3. The first bullet point in the “Highlights” section (not “Highlight”) is a good example. There are at least two common English errors in that sentence. There are more throughout the abstract and introduction.

Reponses:

Thank you very much for your suggestions. Considering that this journal does not need “Highlights” section, items 1, 2 and 3 have been deleted.

In addition, as you suggested, we have revised the abstract and introduction with yellow color highlights. (Line 15-32; Line 36-50; Line 63-89)

Specific technical comments:

Q4. How many wild type (WT) and transparent mutants were used to extract RNA from? What were the overall numbers of fish used?

Reponses:

Thank you very much for your comments. In the present study, five fry were collected for one replicated group. Three replicated groups for TMs (total 15 carps) and WTs (total 15 carps) were all used for RNA extraction, respectively (Line160-163).

Q5. Under section 2.6: How were correlation analyses carried out? (i.e. what statistical test was used to establish significant correlations?). This section needs a lot of further clarification. This is referred to again in section 3.4 but the methodology for correlation (positive and negative) is unclear. What was significant?

Reponses:

Thank you very much for your comments. Firstly, I'm so sorry that I put Figure 6 in the position of Figure 7 by mistake. We have replaced it with the new Figure 7 in the revised manuscript. In the new Figure 7, these statements for miRNA and target genes were all apparent.

Secondly, as you suggested, the further clarification for the section 2.6 has been briefly described in the revised manuscript as follows. (Line150-161)

“In order to define all the possible miRNA-mRNA interactions, we constructed the DE genes pool (p<0.05 and |log2(Fold change)|≥1) and DE miRNAs target genes pool (p-adjust<0.05 and |log2(Fold change)|≥1), respectively. Then the interaction analysis was carried out in the two gene pools. Briefly, the target genes of the DE miRNAs were firstly predicted basing on the miRanda (3.3a). Then, overlapped with the DE genes obtained by RNA-Seq, the target DE genes of DE miRNAs were further detected. According to the expression trend, the relationship between DE miRNAs and their corresponding target DE genes was determined to be up-up, up-down, down-down and down-up, respectively. The miRNA-mRNA regulatory networks were visualized by using Cytoscape software [30]. Given that miRNAs could negatively regulate the expression of their target mRNAs in most cases, only the data with negative relationship (up-down and down-up) between miRNA and mRNA expression were selected for the miRNA-mRNA network mapping in the present study.”

Round 2

Reviewer 1 Report

Overall, most of my questions have been answered and concerns have been addressed. However, the following two problems should be adressed:

Have you checked the reliability of your housekeeping genes? e.g. using qBase? Why did you choose a different cut-off for up- and down-regulated Genes? Why not the same? The cut-off should be the same, otherwise the statement will be consumed. Why are the genes with FC between 3 and 15 not included in the analysed down-regulated DEGs?

Author Response

Comments and Suggestions for Authors

Overall, most of my questions have been answered and concerns have been addressed. However, the following two problems should be addressed:

Have you checked the reliability of your housekeeping genes? e.g. using qBase? Why did you choose a different cut-off for up- and down-regulated Genes? Why not the same? The cut-off should be the same, otherwise the statement will be consumed. Why are the genes with FC between 3 and 15 not included in the analysed down-regulated DEGs?

Responses:

Thank you very much for your help with the paper review. The responses to your comments and answers to specific questions are listed online. To facilitate the paper review, modification in the revised manuscript related to the comments/questions raised have been highlighted in yellow. Thanks a lot again.

(1) Have you checked the reliability of your housekeeping genes? e.g. using qBase?

Responses:

Thank you very much for your comments.

In the present study, the primers of the housekeeping genes were referenced from others published article about common carp (Zhang, Z.; Zheng, Z.; Cai, J.; Liu, Q.; Yang, J.; Gong, Y.; Wu, M.; Shen, Q.; Xu, S. Effect of cadmium on oxidative stress and immune function of common carp (Cyprinus carpio L.) by transcriptome analysis. Aquatic Toxicology. 2017, 192, 171-177.).

In addition, equivalent total RNA from each sample was used for reverse transcription to ensure the consistent amount of cDNA templates used for RT-qPCR (Line 166), and the expression levels of β-actin and Gapdh in WTs and TMs were similar between WTs and TMs.

These two points above suggest the reliability of housekeeping genes.

(2) Why did you choose a different cut-off for up- and down-regulated Genes? Why not the same? The cut-off should be the same, otherwise the statement will be consumed. Why are the genes with FC between 3 and 15 not included in the analysed down-regulated DEGs?

Responses:

Thank you very much for your comments. In this study, the threshold for DE genes were defined with p<0.05 and |log2(Fold change)|≥1 (Line 136). Compared to WT, a total of 331 DE genes were up-regulated while 792 DE genes were down-regulated in TM (Supplemental Table 5). Due to the large amount of data, we only selected top 57 up-regulated DE genes and top 60 down-regulated DE genes in the tables. For the top 57 up-regulated DE genes, the cut-off was FCTM/WT>3, and for the top 60 down-regulated DE genes, the cut-off was FCWT/TM>15.